# Nomogram to Predict the Overall Survival of Colorectal Cancer Patients: A Multicenter National Study

**DOI:** 10.3390/ijerph18157734

**Published:** 2021-07-21

**Authors:** Nasrin Borumandnia, Hassan Doosti, Amirhossein Jalali, Soheila Khodakarim, Jamshid Yazdani Charati, Mohamad Amin Pourhoseingholi, Atefeh Talebi, Shahram Agah

**Affiliations:** 1Urology and Nephrology Research Center, Shahid Beheshti University of Medical Sciences, Tehran 1666663111, Iran; nasrin.borumand@sbmu.ac.ir; 2Department of Mathematics and Statistics, Macquarie University, Sydney, NSW 2109, Australia; hassan.doosti@mq.edu.au; 3School of Mathematical Sciences, University College Cork, T12 XF62 Cork, Ireland; amir.jalali@ucc.ie; 4Department of Biostatistics, School of Medicine, Shiraz University of Medical Sciences, Shiraz 7188614228, Iran; skhodakarim@sbmu.ac.ir; 5Health Sciences Research Center, Biostatistics Department, Addiction Institute, School of Public Health, Mazandaran University of Medical Sciences, Sari 1353447416, Iran; jyazdani@mazums.ac.ir; 6Gastroenterology and Liver Diseases Research Center, Research Institute for Gastroenterology and Liver Disease, Shahid Beheshti University of Medical Sciences, Tehran 1985717413, Iran; pourhoseingholi@sbmu.ac.ir; 7Colorectal Research Center, Iran University of Medical Center, Tehran 1445613131, Iran

**Keywords:** colorectal cancer, cox proportional hazards, nomogram, overall survival, risk factors

## Abstract

Background: Colorectal cancer (CRC) is the third foremost cause of cancer-related death and the fourth most commonly diagnosed cancer globally. The study aimed to evaluate the survival predictors using the Cox Proportional Hazards (CPH) and established a novel nomogram to predict the Overall Survival (OS) of the CRC patients. Materials and methods: A historical cohort study, included 1868 patients with CRC, was performed using medical records gathered from Iran’s three tertiary colorectal referral centers from 2006 to 2019. Two datasets were considered as train set and one set as the test set. First, the most significant prognostic risk factors on survival were selected using univariable CPH. Then, independent prognostic factors were identified to construct a nomogram using the multivariable CPH regression model. The nomogram performance was assessed by the concordance index (C-index) and the time-dependent area under the ROC curve. Results: The age of patients, body mass index (BMI), family history, tumor grading, tumor stage, primary site, diabetes history, T stage, N stage, and type of treatment were considered as significant predictors of CRC patients in univariable CPH model (*p* < 0.2). The multivariable CPH model revealed that BMI, family history, grade and tumor stage were significant (*p* < 0.05). The C-index in the train data was 0.692 (95% CI, 0.650–0.734), as well as 0.627 (0.670, 0.686) in the test data. Conclusion: We improved a novel nomogram diagram according to factors for predicting OS in CRC patients, which could assist clinical decision-making and prognosis predictions in patients with CRC.

## 1. Introduction

According to GLOBOCAN 2020 data, the CRC have been regarded as the fourth most commonly diagnosed cancer globally [1]. In the USA, patients with CRC have reported about 130,000 cases with over 50,000 death records [2]. In European Union countries, CRC is the second common cause of death in the European Union, with 215,000 cases, and second common cancer sites, with 447,000 cases [3]. In Singapore, the CRC is the top rank of cancer and second in the cause of cancer death [4]. According to the cancer registry program in Iran, CRC is considered the third most common cancer in Iran, following only breast and stomach cancer [5,6,7]. The CRC is the fourth most commonly diagnosed cancer in the Iranian male’s population, and the second in females, respectively [1,8,9,10]. Although a great number of investigations have revealed a remarkable variability around the world, and almost 60% of cases happen in developed countries, its overall incidence rate illustrates a slow trend but steady increase (approximately 2% per year) in developed nations. On the contrary, in developing societies and a significant number of Asian countries, the annual incidence is unfortunately anticipated to rise during the next two decades [11].

Nomogram is a simple graphical representation of a statistical prediction model that generates a numerical probability of a clinical event and has been recently applied in prognosis-associated clinical studies with comparable results [12,13,14]. In other words, nomograms which include the histology, tumor grading, history of polyp, the number of involved lymph nodes can be clinically used to predict survival among patients with CRC [15,16].

Many studies have done statistical analysis, including logistic regression or the CPH model, in CRC patients [17,18]. Several studies have implemented survival analysis, including frailty, time-varying Cox, and Cure models in CRC [19,20,21]. Other researchers have presented nomograms, which are the graphical approach to more intuitive perception, in CRC patients [22,23,24,25].

To the best of our knowledge, this study was the first viewpoint of nomogram visualization on the predictive and prognostic factors regarding and OS for CRC in Iran. Also, this is the first Iranian multicenter study that surveys demographic and clinical traits of patients with CRC. The large sample size (*n* = 1868) confirms a vast range of relationships with sufficient statistical analysis power in both train and test sets.

With such a large sample size population, the goal of the historical cohort study was to apply Cox regression to assess the influence of significant factors on CRC patients’ survival rate who registered at three tertiary referral centers in Iran between 2006 and 2019. Then, the nomogram was drawn to generate the probability of survival in CRC patients. The C-index was used for the validation of train and test datasets.

## 2. Materials and Methods

In the study, we gathered both demographic information and clinical characteristics of 1868 patients diagnosed with CRC and referred to three tertiary Hospitals of Iran from 2006 to 2019. Patients in Shahid Faghihi Hospital in Shiraz and Taleghani Hospital in Tehran were considered train sets, and patients in Imam Khomeini Hospital in Mazandaran were applied as the test set. 

The response variable was the time (months) elapsed from the cancer diagnosis until death. Several important clinical factors were included in the model, such as tumor size, the number of involved lymph nodes, distant metastasis, histology, type of treatment, history of polyp and CRC, comorbidity colon diseases (inflammatory bowel disease and irritable bowel syndrome), Diabetes Mellitus, tumor stage, location of the tumor, and demographic variables such as sex, age, education level, smoking and alcohol consumption status, marital status, and BMI. Also, there are some missing data among variables. Patients who had a history of colorectal surgery for any reason except colorectal cancer were excluded. The Ethics Committee of the Iran University of Medical Sciences approved the project (IR.IUMS.REC.1399.1223). The Transparent Reporting of a multivariable prediction model for Individual Prognosis or Diagnosis (TRIPOD), a statement including a 22-item checklist, which aims to improve the reporting of studies developing, validating, or updating a prediction model, whether for diagnostic or prognostic purposes, has been presented in the Appendix A. Figure 1 shows the flowchart of choosing patients in both training and testing sets.

### Statistical Analysis

The participants’ clinical features were represented by reporting the mean with SD for continuous measures and frequency with proportions for categorical ones. The univariable CPH model was implemented to evaluate the effect of some essential factors on CRC patients. Those variables which had *p* < 0.2 in the univariable analysis were candidates for the multivariable regression analysis. The result of the multivariable Cox model was presented as a nomogram. To assess the model performance, concordance index (C-index) and the time-dependent AUC (Area Under the ROC Curve) at different time points were calculated.

The significance level for the statistical analysis was considered 0.05. The R 4.1.0 software (http://www.r-project.org) with the survival and rms packages was applied for statistical analysis. Also, the DynNom package was used to construct the dynamic nomogram [26].

## 3. Results

A total of 1649 CRC patients, including Shiraz and Tehran cities, were included in the study as the train set. Also, another dataset from Mazandaran was applied as the test set (*n* = 219). Overall, 59.7% (*n* = 988) were male and 40.3% (*n* = 666) were female. The median follow-up time was 21.86 months (IQR: 9–37.2 and range 1, 179 months). The mean (SD) age of patients was 54 (14) years; moreover, the detailed demographic and clinical characteristics of all the CRC patients, according to survival status, were summarized in Table 1. 

In this regard, factors associated with survival are listed in Table 1 based on the univariable Cox regression. The table revealed that age, BMI, family history, tumor grade, stage of the tumor, primary site, Diabetes history, T stage, N stage, and types of treatment are significant in the univariable Cox model. Those variables with *p* < 0.2 in the univariable analysis were incorporated in the multivariable Cox model given in Table 2. 

The multivariable Cox model’s output presented that BMI, family history, grade tumor, and tumor stage are statistically significant (*p* < 0.05).

The HR of death for patients with BMI < 18 (underweight) is 94% more than those with overweight persons, which was significant (HR = 1.94, *p* < 0.05). Also, the HR in normal-weight persons is 42% more than the overweight persons (HR = 1.42, *p* < 0.05). The HR in patients who do not have a family history of cancer is 42% less than those who do not have a family history (HR = 0.58, *p* = 0.002).

An HR of tumor grade categories indicated that both moderate and poor differentiation had worse prognoses than poorly differentiated (HR = 1.5; HR = 2.67, *p* < 0.05).

By worsening the tumor stage, the HR is increased significantly in CRC patients. That means the higher the stage of the tumor, the higher the HR. The HR in patients with stage IV of CRC is about 3.2 times more than stage I of patients (HR = 3.24, *p* = 0.005).

Based on the results of multivariable analysis, we established a dynamic web-based nomogram to calculate the survival probability (Dynamic Nomogram (shinyapps.io), https://nbshiny.shinyapps.io/DynNomColorectal/). Using it, one can predict the long-term survival of patients with CRC (Figure 2). This statistic tool that combines all prognostic indexes represents a graphical model that simply calculates the individualized overall survival probability for CRC patients.

### Validation of Nomogram

The C-index for the nomogram was calculated for train and test datasets. The C-index in the train set was 0.692 (95% CI, 0.650–0.734). The demographic and clinical characteristics of all the CRC patients of the test set, according to survival status, were summarized in Table 3. Also, the C-index of the test set was estimated as 0.627 (0.670, 0.686), which showed the nomogram provided good discernment. 

In addition, to assess the model performance internally, the time-dependent AUC was calculated at different time points. The results have been presented in Figure 3.

## 4. Discussion

In the present study, the univariable and multivariable Cox regression models were applied, and then the nomogram diagram was constructed to predict OS, which was able to provide individualized estimates of potential survival benefits. The significant factors of the study are the BMI, family history of cancer, histology, depth of invasion. The C-index of the train and test dataset was estimated at 0.692 and 0.627, respectively. Also, time-dependent AUC was evaluated at separate times.

A significant number of modeling techniques in survival analysis have been suggested for proportional hazard and non-proportional hazards [20,27,28]. Their results of the Cox model showed that tumor size and grade of tumor are vital in the survival of CRC patients. Similar to our study, previous surveys have reported the relationship between age at diagnosis and the 5-year survival [21]. Zhao et al. (2020) applied machine learning to predict OS more accurately in colon cancer patients and presented the predictive model in nomograms for patients and clinicians [21]. They also used the Cox regression model to find the predictive factors on cancer. Some variables such as age, highest CEA level, the primary site of a tumor, treatment type, and the number of involved lymph nodes were significant. In our study, we did not have the CEA level; moreover, the number of involved lymph nodes and types of treatment were not statistically significant.

Our study’s critical result revealed a significant relationship between the survival of CRC patients with marital status, consistent with Zhang et al. study [29]. In their study, sex, race, CEA status, tumor size, tumor site, marital status, histology, grade and tumor stage, the extent of surgery, and metastasis were considered significant prognostic factors of CRC. In our study, histology, grade, and tumor stage are significant, which were compatible with their study [29,30,31].

Li et al. showed the age of patients, sex, depth of invasion, and tumor location were significant prognostic factors [32]. In the study, the C-indexes of the nomogram for the prediction of OS were 0.723 and 0.716 in the training and testing group, respectively. In another survey, tumor size and involved lymph nodes were substantial, while these variables were not significant prognostic factors in Yu’s study [14]. In our study, the C-index of the train and test sets was estimated at 0.692 and 0.627.

Similar to our results, Li et al. showed that tumor size and the number of involved lymph nodes were significant prognostic factors in CRC [33]. In their study, several serum tumor biomarkers, including CA19-9, CA242, CA72-4, CA50, and CA125 were studied in association with prognosis. They used the univariable and multivariable Cox regression models to evaluate the relationship between these markers and survival outcomes. They also draw the nomograms based on multivariable Cox regression model analysis for OS. Also the C-indexes of their study were 0.772 and 0.715. In our investigation, the number of involved lymph nodes was significant in the univariable Cox regression model, but the variable was not considered as the main factor in multivariable CPH. 

A survey has revealed that age, depth of invasion, number of involved lymph nodes, treatment type were significant in CRC, consistent with our study [34]. The univariable and multivariable Cox analyses were conducted to predict the individual risk of metachronous peritoneal carcinomatosis after surgery for non-metastatic CRC. The depth of invasion and pathology of primary tumors have been identified as risk factors for CRC patients’ survival, which are compatible with our study. In their study, the C-index in both train and test datasets were 80% and 70%, while in our study, these values were 0.692 and 0.627. 

Li et al. have performed survival analysis to assess an effective prognostic model for predicting survival in resected colorectal cancer patients [18]. They applied multivariable Cox regression analysis to identify significant prognostic. Their results demonstrated that age, CEA level, the number of involved lymph nodes, tumor stage, histological type, tumor grading, tumor location, treatment type, and lymph-vascular invasion were significant. In our study, the stage and grade of cancer were significant, which was consistent with the findings of treatment in the study of Li et al.

### Strengths and Limitations

The first key strength of the present survey is the large sample size of a multi-center study together with a small number of missing data. The second fundamental strength of this study is the long-term follow-up period. The limitation of the study is that some indispensable factors such as CEA level, Albumin, and Fibrinogen levels have not been recorded in the patients ‘questionnaire.

## 5. Conclusions

Our research investigated a historical cohort of 1868 CRC patients to create a web based nomogram using both demographic and clinical features to predict OS. The nomograms can act as a visual tool to integrate clinical characteristics to predict individualized cancer prognoses.

## Figures and Tables

**Figure 1 ijerph-18-07734-f001:**
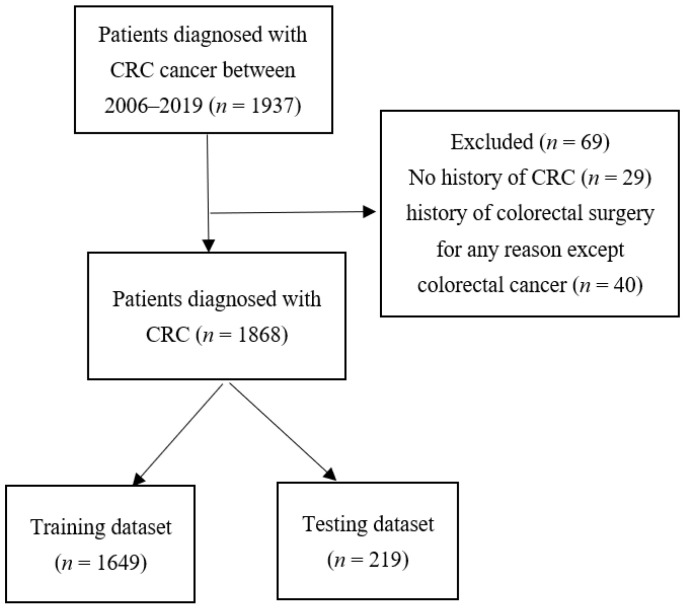
Flow chart of the patient selection process.

**Figure 2 ijerph-18-07734-f002:**
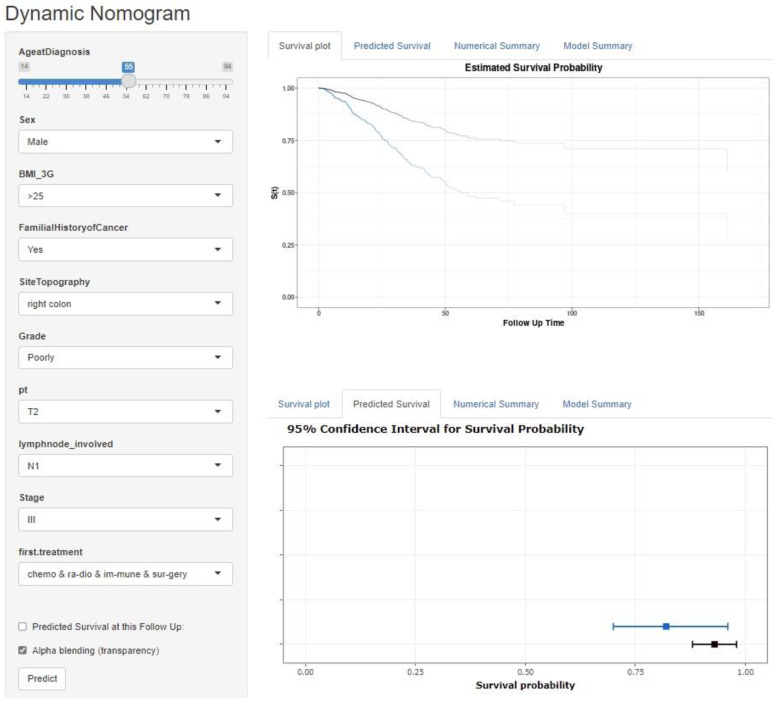
Dynamic nomogram for the Cox proportional hazards model, fitted to the Colorectal cancer patient’s data, on web page (Dynamic Nomogram (shinyapps.io), https://nbshiny.shinyapps.io/DynNomColorectal/). The Kaplan-Meier plots display survival curve correspond to 55 years old male, BMI > 25, have a family history, cancer in the right colon, T2 T-stage, N1 N-stage, stage III, receive all treatments, and well-differentiated grade (in black color) vs. a patient with the same characteristics and poorly differentiated grade (in blue color), shown in the left side of the picture (upper). The patients’ corresponding predicted survival probability and 95% confidence intervals at a specific time is given in the ‘Predicted survival’ tab, shown on the left side of the picture (lower). The predicted value with corresponding confidence interval and the formatted model output summary are presented in the ‘Numerical Summary’ and ‘Model Summary’ tabs, respectively.

**Figure 3 ijerph-18-07734-f003:**
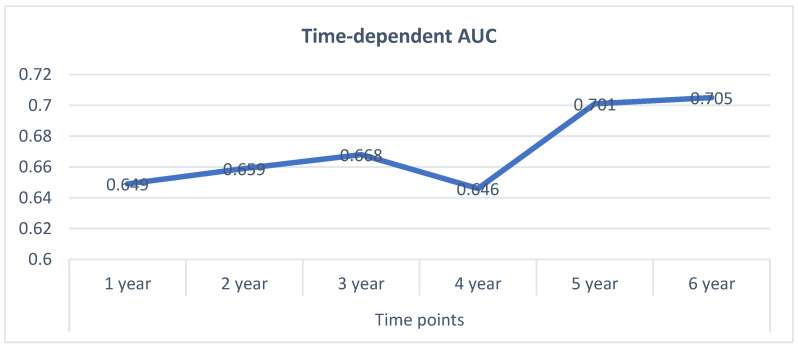
Time-dependent AUC values for internal nomogram validation.

**Table 1 ijerph-18-07734-t001:** Demographic and clinical characteristics; univariable Cox model results to explore the effect of demographic and clinical characteristics variables on survival among patients with CRC.

Variables		Status			
		Alive (*n* = 1265)	Dead (*n* = 384)	HR (CI)	*p*-Value
Follow up duration; median (IQR)		23.50 (8.87–38.93)	18.00 (9.73–32.23)	--	
Age at diagnosis (years); mean (SD)		55.53 (14.09)	55.02 (15.22)	1.00 (99, 1.01)	0.043
Tumor size (cm); mean (SD)		4.59 (2.99)	4.66 (3.12)	1.01 (0.97, 1.04)	0.574
Sex	Male	749 (76.0%)	236 (24.0)	1.12 (0.91, 1.38)	0.256
	Female	516 (77.7%)	148 (22.3%)	-	-
BMI	<18	72 (59.0%)	50 (41.0%)	2.44 (1.72, 3.47)	<0.001
	18–25	490 (76.2%)	153 (23.8%)	1.67 (1.27, 2.18)	<0.001
	>25	402 (82.5%)	85 (17.5%)	-	-
Smoking	No	867 (77.1%)	258 (22.9%)	0.962 (0.77, 1.21)	0.737
	Yes	339 (76.4%)	105 (23.6%)	-	-
Diabetes Mellitus	No	985 (77.0%)	294 (23.0%)	0.77 (0.49, 1.20)	0.250
	Yes	108 (83.7%)	21 (16.3%)	-	-
Family history	No	756 (74.7%)	256 (25.3%)	0.736 (0.59, 0.92)	0.006
	Yes	466 (79.9)	117 (20.1%)	-	-
Primary site	Right colon	338 (79.0)	90 (21.0%)	-	-
	Left colon	737 (78.7)	199 (21.3%)	1.07 (0.083, 1.37)	0.593
	Rectum	107 (66.9%)	53 (33.1%)	1.35 (0.95, 1.90)	0.085
	Transverse	3 (75.0%)	1 (25.0%)	1.27 (0.17, 9.19)	0.807
Tumor grade	Well-differentiated	593 (81.9%)	131 (18.1%)	-	-
	Moderately-differentiated	295 (76.4%)	91 (23.6%)	1.41 (1.08, 1.85)	0.012
	Poorly-differentiated	68 (63.0%)	40 (37.0%)	2.22 (1.56, 3.17)	<0.001
T stage	T0	636 (81.7%)	142 (18.3%)	-	-
	T1	161 (71.9%)	63 (28.1%)	1.47 (1.09, 1.99)	0.010
	T2	100 (78.7%)	27 (21.3%)	1.01 (0.66, 1.54)	0.944
	T3	178 (71.2%)	72 (28.8%)	1.49 (1.12, 1.99)	0.006
	T4	9 (47.4%)	10 (52.6%)	2.70 (1.42, 5.14)	0.002
N stage	N0	606 (79.3%)	158 (20.7%)	-	-
	N1	389 (76.7%)	118 (23.3%)	1.32 (1.04, 1.68)	0.022
	N2	23 (59.0%)	16 (41.0%)	2.31 (1.38, 3.88)	0.001
Stage of tumor	I	168 (84.4%)	31 (15.6%)	-	-
	II	410 (79.8%)	104 (20.2%)	1.27 (0.84, 1.91)	0.248
	III	374 (77.3%)	110 (22.7%)	1.69 (1.13, 2.54)	0.010
	IV	129 (65.8%)	67 (34.2%)	2.8 (1.61, 3.82)	<0.001
Types of treatment	Surgery	948 (79.0%)	252 (21.0%)	-	-
	Chemotherapy, radiotherapy, immunotherapy, surgery	203 (66.1%)	104 (33.0%)	1.38 (1.10, 1.74)	0.005

**Table 2 ijerph-18-07734-t002:** Multivariable Cox model results to explore the effect of Factors associated with survival among patients with CRC.

Variables		HR(CI)	*p*-Value
Age at diagnosis (years)		1.006 (0.99, 1.01)	0.098
Sex	Male	-	-
	Female	1.15 (0.85, 1.56)	0.351
BMI	<18	1.94 (1.21, 3.12)	0.006
	18–25	1.42 (1.00, 2.01)	0.045
	>25	-	-
Family history	No	0.58 (0.42, 0.82)	0.002
	Yes	-	-
Primary site	Right colon		
	Left colon	0.87 (0.57, 1.34)	0.551
	Rectum	1.24 (0.80, 1.91)	0.332
	Transverse	1.43 (0.19, 10.67)	0.726
Tumor grade	Well-differentiated	-	-
	Moderately-differentiated	1.50 (1.07, 2.10)	0.019
	Poorly-differentiated	2.67 (1.69, 4.21)	<0.001
T stage	T0	-	-
	T1	0.87 (0.49, 1.55)	0.653
	T2	1.02 (0.53, 1.97)	0.935
	T3	1.16 (0.71, 1.89)	0.545
	T4	1.32 (0.55, 3.17)	0.533
N stage	N0	-	-
	N1	1.51 (0.78, 2.92)	0.213
	N2	2.17 (0.93, 5.04)	0.070
Stage of tumor	I	-	-
	II	1.16 (0.64, 2.12)	0.610
	III	1.06 (0.46, 2.47)	0.877
	IV	3.24 (1.42, 7.41)	0.005
Types of treatment	Surgery	-	-
	Chemotherapy, radiotherapy, immunotherapy, surgery	1.22 (0.83, 1.78)	0.310

**Table 3 ijerph-18-07734-t003:** Demographic and clinical characteristics of CRC cancers in train dataset.

Variables		Alive (*n* = 111)	Dead (*n* = 108)
Follow up duration; median (IQR)		55.0 (37.0–70.0)	26.5 (13.5–42.5)
Age at diagnosis (years); mean (SD)		58.0 (15.2)	60.5 (15.4)
Sex	Male	63 (50.8%)	61 (49.2%)
	Female	48 (50.5%)	47 (49.5%)
BMI	<18	6 (60.0%)	4 (40.0%)
	18–25	50 (50.5%)	49 (49.5%)
	>25	55 (50.0%)	55 (50.0%)
Family history	No	75 (49.3%)	77 (50.7%)
	Yes	36 (53.7%)	31 (46.3%)
Primary site	Right colon	61 (55.0%)	50 (45.0%)
	Left colon	41 (46.1%)	48 (53.9%)
	Rectum	0 (0.0%)	0 (0.0%)
	Transverse	8 (50.0%)	8 (50.0%)
Tumor grade	Well-differentiated	44 (57.1%)	33 (42.9%)
	Moderately-differentiated	63 (50.4%)	62 (49.6%)
	Poorly-differentiated	4 (23.5%)	13 (76.5%)
T stage	T0	63 (46.7%)	72 (53.3%)
	T1	14 (43.8%)	18 (56.3%)
	T2	15 (83.3%)	3 (16.7%)
	T3	14 (58.3%)	10 (41.7%)
	T4	5 (50.0%)	5 (50.0%)
N stage	N0	0 (0.0%)	0 (0.0%)
	N1	73 (57.5%)	54 (42.5%)
	N2	38 (41.3%)	54 (58.7%)
Stage of tumor	I	24 (77.4%)	7 (22.6%)
	II	51 (65.4%)	27 (34.6%)
	III	32 (46.4%)	37 (53.6%)
	IV	4 (9.8%)	37 (90.2%)
Types of treatment	Surgery	7 (46.7%)	8 (53.3%)
	Chemotherapy, radiotherapy, immunotherapy, surgery	104 (51.0%)	100 (49.0%)

## Data Availability

The datasets used and analyzed during the current study are available from the corresponding author on reasonable request.

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
