# Peer review of "Nomogram to Predict the Overall Survival of Colorectal Cancer Patients: A Multicenter National Study"

_ijerph, 2021, doi:10.3390/ijerph18157734_

Round 1
Reviewer 1 Report
- “Colorectal cancer (CRC) is the third foremost cause of cancer-related death and the fourth most commonly diagnosed cancer globally” Please confirm it.
- “This study was the first viewpoint on the predictive and prognostic factors regarding OS for CRC to the best of our knowledge”. There have been many articles on predicting OS based on nomogram in CRC.
- “this is the first multicenter study that surveys both demographic and clinical traits of patients with CRC among CRC patients.” There have been many multicenter studies on predicting OS in CRC.
- What were the inclusion and exclusion criteria for the patients included in this study? Please show the flow chart for inclusion of patients.
- “Those variables which had P < 0.2 in the univariable analysis were candidates for the multivariable regression analysis.” What is the basis?
- 1054+ 688 ≠1774. Is there an error in the statistical process?
- The sum of the data in Table 1 does not match, such as 453+ 602≠1054. Table 1 needs to be double-checked by the author.
- The TRIPOD (Transparent Reporting of a multivariable prediction model for Individual Prognosis Or Diagnosis) Statement includes a 22-item checklist, which aims to improve the reporting of studies developing, validating, or updating a prediction model, whether for diagnostic or prognostic purposes. The author should report each item as required and include the checklist as an attachment.
- The authors should have increased the median survival time for each group of patients in table1, rather than just the number of patients who survived or died.
- Why is the HR for stage II III IV less than 1? Please provide a reasonable explanation.
- Is tumor stage a clinical stage or a pathological stage? Sur & chemo is chemotherapy followed by surgery?
- Why are there patients in stages 2 and 3 who receive chemotherapy or radiotherapy alone? Because all patients in stages 1-3 can be treated with surgery. This is contrary to current treatment principles. Could the authors please show the number of patients receiving each treatment and explain?
- Why do patients with stage 4 receive only radiotherapy or surgery? What type of surgery is the surgery? Radical surgery? Palliative surgery? No post-operative chemotherapy? Stage 4 patients are predominantly treated systemically. This is contrary to current treatment principles and the authors are asked to show the number of specific patients and provide a reasonable explanation.
- How do I use Nomogram? Please give an example. Why is there no stage1 score? The Nomogram is too small for many details to be seen.
- This paper lacks validation of the Nomogram and suggests that the authors use a third centre for validation of the data. Alternatively, a random sample of patients could be selected for validation.
- This paper lacks an evaluation of Nomogram. ROC curves, DCA curves, and calibration curves are suggested for evaluation. This is one of the main drawbacks of this paper.
- This paper lacks a comparison of this nomogram with other existing nomograms, including a comparison of the AUC and a comparison of the goodness of fit. This is the main drawback of this paper.
- The English and logic of this paper need to be further improved.
Author Response
Reviewer #1
Open Review
(x) I would not like to sign my review report
( ) I would like to sign my review report
English language and style
(x) Extensive editing of English language and style required
( ) Moderate English changes required
( ) English language and style are fine/minor spell check required
( ) I don't feel qualified to judge about the English language and style
|
Yes |
Can be improved |
Must be improved |
Not applicable |
|
|
Does the introduction provide sufficient background and include all relevant references? |
( ) |
( ) |
(x) |
( ) |
|
Is the research design appropriate? |
( ) |
( ) |
(x) |
( ) |
|
Are the methods adequately described? |
( ) |
( ) |
(x) |
( ) |
|
Are the results clearly presented? |
( ) |
( ) |
(x) |
( ) |
|
Are the conclusions supported by the results? |
( ) |
( ) |
(x) |
( ) |
Comments and Suggestions for Authors
- “Colorectal cancer (CRC) is the third foremost cause of cancer-related death and the fourth most commonly diagnosed cancer globally” Please confirm it.
Answer 1: Thank you for your attention. A reference was added in line 57.
- “This study was the first viewpoint on the predictive and prognostic factors regarding OS for CRC to the best of our knowledge”. There have been many articles on predicting OS based on nomogram in CRC.
Answer 2: Thank you for your consideration. A multicenter study of CRC patients based on Nomogram in Iran have not done until now. The sentence was corrected in line 77-78.
- “This is the first multicenter study that surveys both demographic and clinical traits of patients with CRC among CRC patients.” There have been many multicenter studies on predicting OS in CRC.
Answer 3: I appreciate your great comment. This is the first multicenter study among Iranian patients. The sentence was edited in line 76-77.
- What were the inclusion and exclusion criteria for the patients included in this study? Please show the flow chart for inclusion of patients.
Answer 4: Many thanks for your attention. Please kindly find the manuscript file in line 99- 118. The flow chart is drawn in the figure as follow:
- “Those variables which had P < 0.2 in the univariable analysis were candidates for the multivariable regression analysis.” What is the basis?
Answer 5: Considering that more traditional levels such as 0.05 can fail in identifying variables known to be important, based on following references:
- Bendel RB, Afifi AA. Comparison of stopping rules in forward regression. Journal of the American Statistical Association. 1977;72:46–53. doi: 10.2307/2286904.
- Mickey J, Greenland S. A study of the impact of confounder-selection criteria on effect estimation. American Journal of Epidemiology. 1989;129:125–137.],
so, the following reference [Applied Survival Analysis: Regression Modeling of Time-to-Event Data Book by David W. Hosmer, Stanley Lemeshow, and Susanne May] recommend that variables which had P < 0.2 in the univariable analysis were candidates for the multivariable regression analysis.
In addition, the bivariate insignificant variable can turn up statistical significance in multiple regression. It's called a suppressor effect.
- 1054+ 688 ≠1774. Is there an error in the statistical process?
Answer 6: There is not a result error. In fact, this mismatch is due to missing data in some variables. The total number of registered data was 1868 (1649 in train dataset and 219 in test dataset), and there was missing related to some variables. Therefore, the sum of frequency may not be equal to the total number and maybe slightly less.
- The sum of the data in Table 1 does not match, such as 453+ 602≠1054. Table 1 needs to be double-checked by the author.
Answer 7: I appreciate your great comment. We examined the validation of data and revised the table 1 carefully. Please consider that due to missing data in some variables, the sum of frequency may not be equal to the total number and maybe slightly less. Please find the line 142.
- The TRIPOD (Transparent Reporting of a multivariable prediction model for Individual Prognosis Or Diagnosis) Statement includes a 22-item checklist, which aims to improve the reporting of studies developing, validating, or updating a prediction model, whether for diagnostic or prognostic purposes. The author should report each item as required and include the checklist as an attachment.
Answer 8: In this way, we add the checklist as an attachment in supplementary 1. Please kindly see the Supplementary file.
- The authors should have increased the median survival time for each group of patients in table1, rather than just the number of patients who survived or died.
Answer 9: Thanks for your kind attention. The median and range of follow up time was added in the table 1. Please see the line 143.
- Why is the HR for stage II III IV less than 1? Please provide a reasonable explanation.
Answer 10: Many thanks for your consideration. We examined the validation of data and revised the table 1 carefully. Since we omit one dataset and add two other datasets, the outputs change. Please kindly find the new results in Table 1, line 143.
- Is tumor stage a clinical stage or a pathological stage? Sur & chemo is chemotherapy followed by surgery?
Answer 11: Many thanks for your attention. We examined the validation of data and revised the table 1 carefully. Since we omit one dataset and add two other datasets, the outputs change. Please kindly find the new results in Table 1, line 143.
- Why are there patients in stages 2 and 3 who receive chemotherapy or radiotherapy alone? Because all patients in stages 1-3 can be treated with surgery. This is contrary to current treatment principles. Could the authors please show the number of patients receiving each treatment and explain?
Answer 12: Thanks for your great consideration. We examined the validation of data and revised the table 1 carefully. Since we omit one dataset and add two other datasets, the outputs change. Please kindly find the new results in Table 1, line 143.
- Why do patients with stage 4 receive only radiotherapy or surgery? What type of surgery is the surgery? Radical surgery? Palliative surgery? No post-operative chemotherapy? Stage 4 patients are predominantly treated systemically. This is contrary to current treatment principles and the authors are asked to show the number of specific patients and provide a reasonable explanation.
Answer 13: Many thanks for your attention. We examined the validation of data and revised the table 1 carefully. Since we omit one dataset and add two other datasets, the outputs change. Please kindly find the new results in Table 1, line 143.
- How do I use Nomogram? Please give an example. Why is there no stage1 score? The Nomogram is too small for many details to be seen.
Answer 14: We tried to change the format of Nomogram to be user-friendly. For this purpose, we used the dynamic nomogram, presented in Figure 1. The relevant nomogram chart is available at the following web address. In this chart format, you can select the patient profile, then determine the year of survival, and selecting the prediction option, the probability of survival will be calculated and appeared in the predicted survival window. An image of dynamic Nomogram figure has also been added to the manuscript.
We established a dynamic web-based nomogram to calculate the survival probability (Dynamic Nomogram (shinyapps.io), https://nbshiny.shinyapps.io/DynNomColorectal/)
Please see the line 187.
- This paper lacks validation of the Nomogram and suggests that the authors use a third centre for validation of the data. Alternatively, a random sample of patients could be selected for validation.
Answer 15: Thanks for your helpful comment. In this way, we selected a center as an external validation and used its data for evaluation as a test set. Also, C index indices were calculated for both train and test datasets. Moreover, the AUC index was calculated for different time points in the data. Please see the line 187-199.
- This paper lacks an evaluation of Nomogram. ROC curves, DCA curves, and calibration curves are suggested for evaluation. This is one of the main drawbacks of this paper.
Answer 16: Thanks for your helpful suggestion. In this regard, we selected a center as an external validation and used its data for evaluation as a test set. Also, C index indices were calculated for both train and test datasets. Moreover, the AUC index was calculated for different time points in the data. Please see the line 187-199.
- This paper lacks a comparison of this nomogram with other existing nomograms, including a comparison of the AUC and a comparison of the goodness of fit. This is the main drawback of this paper.
Answer 17: Many thanks for your helpful advice. Both C-index and AUC were added to the outputs and discussion. Please find the lines 207-210, 232,239,242,249.
- The English and logic of this paper need to be further improved.
Answer 18: Many thanks for your great attention. The manuscript was checked by a native speaker.
Reviewer 2 Report
GENERAL COMMENTS
- In this manuscript, the authors propose a nomogram aimed to predict OS in CRC patients based on a multivariable analysis on 1774 patients.
- Therefore, the analysis was conducted on a large number of patients. Unfortunately, I find it very impractical to use such a complex nomogram, which includes so many parameters, in daily clinical practice. Nowadays, I believe that a more "convenient" solution, especially in very busy departments, is that of an electronic system in which the operator enters the data (stage, age, gender ...) quickly and immediately has an estimate of the prognosis, or better of different outcomes. In any case, the graphics of the nomogram seem unclear to me.
- An independent cohort analyzed for the validation of the nomogram is missing. This aspect is now increasingly required in the design of predictive models.
- Extensive editing of English language and style required.
MINOR COMMENTS
- TITLE: Nomogram for predicting the prognostic value of colorectal cancer patients: A multicenter national study – Please consider to change as follows: “Nomogram for predicting outcome in colorectal cancer patients: A multicenter national study” or something similar.
- To make it easier to read the figures I suggest changing (for example) 300000 to 300,000.
- INTRO – “Also, this is the first multicenter study that surveys both demographic and clinical traits of patients with CRC among CRC patients”; sorry but for me unclear.
- MAT & MET – “A nomogram is a graphical representation of statistical models, such as the Cox proportional hazards model for survival data, which involves several independent variables to predict the survival probability [25].” – the definition of the nomogram was already given in the introduction section.
- RESULTS – “Generally, 59.6% (n = 1058) were male and 40.4% (n = 716) were female” – I suggest changing “Generally” with “Overall”.
- RESULTS – “The mean age of patients was 57 (14) years” – unclear (14)
- TABLE – A legend should explain the abbreviations (like “sur”)
Author Response
Reviewer #2
Open Review
(x) I would not like to sign my review report
( ) I would like to sign my review report
English language and style
(x) Extensive editing of English language and style required
( ) Moderate English changes required
( ) English language and style are fine/minor spell check required
( ) I don't feel qualified to judge about the English language and style
|
Yes |
Can be improved |
Must be improved |
Not applicable |
|
|
Does the introduction provide sufficient background and include all relevant references? |
( ) |
(x) |
( ) |
( ) |
|
Is the research design appropriate? |
( ) |
(x) |
( ) |
( ) |
|
Are the methods adequately described? |
(x) |
( ) |
( ) |
( ) |
|
Are the results clearly presented? |
( ) |
(x) |
( ) |
( ) |
|
Are the conclusions supported by the results? |
( ) |
(x) |
( ) |
( ) |
Comments and Suggestions for Authors
GENERAL COMMENTS
- In this manuscript, the authors propose a nomogram aimed to predict OS in CRC patients based on a multivariable analysis on 1774 patients. Therefore, the analysis was conducted on a large number of patients. Unfortunately, I find it very impractical to use such a complex nomogram, which includes so many parameters, in daily clinical practice. Nowadays, I believe that a more "convenient" solution, especially in very busy departments, is that of an electronic system in which the operator enters the data (stage, age, gender ...) quickly and immediately has an estimate of the prognosis, or better of different outcomes. In any case, the graphics of the nomogram seem unclear to me.
Answer 1: We tried to change the format of Nomogram to be user-friendly. For this purpose, we used the dynamic nomogram, presented in Figure 1. The relevant nomogram chart is available at the following web address. In this chart format, you can select the patient profile, then determine the year of survival, and selecting the prediction option, the probability of survival will be calculated and appeared in the predicted survival window. An image of dynamic Nomogram figure has also been added to the manuscript. we established a dynamic web-based nomogram to calculate the survival probability (Dynamic Nomogram (shinyapps.io),https://nbshiny.shinyapps.io/DynNomColorectal/)
Please see the line 187.
- An independent cohort analyzed for the validation of the nomogram is missing. This aspect is now increasingly required in the design of predictive models.
Answer 2: Many thanks for your constructive comment. We selected a center as an external validation and used its data for evaluation as a test set. Then, C index indices were calculated for both training and test datasets. Also, the AUC index was also calculated for different time points in the data. Please see the line 187-199.
- Extensive editing of English language and style required.
Answer 3: Many thanks for your kind attention. The manuscript was checked by a native speaker.
MINOR COMMENTS
- TITLE: Nomogram for predicting the prognostic value of colorectal cancer patients: A multicenter national study – Please consider to change as follows: “Nomogram for predicting outcome in colorectal cancer patients: A multicenter national study” or something similar.
Answer 4: Many thanks for your kind attention. The name was partially changed.
- To make it easier to read the figures I suggest changing (for example) 300000 to 300,000.
Answer 5: We tried to change the format of Nomogram to be user-friendly. For this purpose, we used the dynamic nomogram, presented in Figure 1. The relevant nomogram chart is available at the following web address. In this chart format, you can select the patient profile, then determine the year of survival, and selecting the prediction option, the probability of survival will be calculated and appeared in the predicted survival window. An image of dynamic Nomogram figure has also been added to the manuscript. We established a dynamic web-based nomogram to calculate the survival probability (Dynamic Nomogram (shinyapps.io), https://nbshiny.shinyapps.io/DynNomColorectal/). Please see line 170.
- INTRO – “Also, this is the first multicenter study that surveys both demographic and clinical traits of patients with CRC among CRC patients”; sorry but for me unclear.
Answer 6: Thanks for your attention. The sentence was corrected. Please find the line76-77.
- MAT & MET – “A nomogram is a graphical representation of statistical models, such as the Cox proportional hazards model for survival data, which involves several independent variables to predict the survival probability [25].” – the definition of the nomogram was already given in the introduction section.
Answer 7: Many thanks for your great attention. It was corrected. Please find lines 126-129.
- RESULTS – “Generally, 59.6% (n = 1058) were male and 40.4% (n = 716) were female” – I suggest changing “Generally” with “Overall”.
Answer 8: Many thanks for your suggestion. Generally was substituted with Overall. Please see line 137.
- RESULTS – “The mean age of patients was 57 (14) years” – unclear (14)
Answer 9: Thanks for your attention. It shows mean(SD) of the variable. Please find the line 139.
- TABLE – A legend should explain the abbreviations (like “sur”)
Answer 10: Many thanks for your consideration. The full name was substituted. Please see line 143,152 in tables 1,2.
Reviewer 3 Report
Borumandnia et al. study Nomogram for predicting the prognostic value of colorectal cancer patients: A multicenter national study is quite interesting . However , following comments must be addressed before considering for publication in IJERPH.
minor comments
- Abberviations are missing makes difficult to read manuscript
- quality of figure is poor must be improved
- while reading the manuscript I found many typo mistakes that must be considered
- proper formatting is also required ....Text is in different formats
Major comments
- G score is missing... how sample size was measured ?
- KM s survival curve analysis , chi square test on multivarite , must be applied
- The historical cohort studies are not good for diseases with a long latency . Second , the differential loss to follow up can introduce bais . Can authors explain it ?
Author Response
Reviewer #3
Open Review
(x) I would not like to sign my review report
( ) I would like to sign my review report
English language and style
( ) Extensive editing of English language and style required
(x) Moderate English changes required
( ) English language and style are fine/minor spell check required
( ) I don't feel qualified to judge about the English language and style
|
Yes |
Can be improved |
Must be improved |
Not applicable |
|
|
Does the introduction provide sufficient background and include all relevant references? |
( ) |
(x) |
( ) |
( ) |
|
Is the research design appropriate? |
( ) |
(x) |
( ) |
( ) |
|
Are the methods adequately described? |
( ) |
(x) |
( ) |
( ) |
|
Are the results clearly presented? |
(x) |
( ) |
( ) |
( ) |
|
Are the conclusions supported by the results? |
(x) |
( ) |
( ) |
( ) |
Comments and Suggestions for Authors
Borumandnia et al. study Nomogram for predicting the prognostic value of colorectal cancer patients: A multicenter national study is quite interesting . However , following comments must be addressed before considering for publication in IJERPH.
minor comments
- Abbreviations are missing makes difficult to read manuscript
Answer 1: Many thanks for your consideration. Abbreviation part was added to the last part of the manuscript. Please see the lines 292-302
- quality of figure is poor must be improved
Answer 2: Many thanks for your suggestion. It was improved and high-quality. Please see line 178.
- while reading the manuscript I found many typo mistakes that must be considered
Answer 3: Thanks for your kind attention. They were corrected by native English language.
- proper formatting is also required. Text is in different formats
Answer 4: Thanks for your comment. The text changed to the same formats.
Major comments
- G score is missing... how sample size was measured?
Answer 5: In this historical cohort multicenter study that is population-based registers, we tried to collect as much sample size as we could to develop and also validate the nomogram. Also, the information for validity has been added.
- KM s survival curve analysis, chi square test on multivarite, must be applied
Answer 6: The univariable and also multivariable analyses Cox model were applied, that are stronger modeling approaches for survival analysis.
- The historical cohort studies are not good for diseases with a long latency. Second, the differential loss to follow up can introduce bias. Can authors explain it?
Answer 7: The specific difficulties relating to survival analysis arise largely from the fact that only some individuals have experienced the event and, subsequently, survival times will be unknown for a subset of the study group. This phenomenon is called censoring and it may arise in the various reasons, for example lost to follow up or a person withdraws during the study period. Due to censoring data, routine methods cannot be used, and special methods of analysis should be applied such as Cox model. Using these modeling, we can handle the bias due to bias of lost to follow up or withdraw during the study period.
Round 2
Reviewer 2 Report
I sincerely congratulate the authors for making a great effort to significantly improve the quality of their manuscript. I have only a few marginal suggestions.
Minor comments:
Title: "Nomogram and predict the overall survival ..." consider to change with "nomogram to predict overall survival"
Abstract (Background):
"Performance was assessed by the concordance index (C-index)" please consider moving this sentence to the material and method paragraph of the abstract section
Author Response
Diane Jiang
Assistant Editor
International Journal of Environmental Research and Public Health
Dear Diane Jiang,
We thank you for the insightful editorial and review comments. We are pleased to be given the opportunity to submit a revised version of our manuscript.
We would like to thank the insightful editorial and reviewer for the constructive and competent criticism. We are very glad to hear that our manuscript has been accepted for publication in the International Journal of Environmental Research and Public Health.
According to the thoughtful comments of reviewers, we perform the reviewers ‘comments. Please find below our point-by-point replies to the reviewers’ comments, which are formatted in bold and numbered, followed by our reply in normal, green-coloured characters (our replies to the comments). Also, all changes in the manuscript are clearly marked by track change. The manuscript has sent in a separate word-processing file (Second revise manuscript with track changes). All authors have read and approved the revised version of the manuscript.
Reviewer #2:
I sincerely congratulate the authors for making a great effort to significantly improve the quality of their manuscript. I have only a few marginal suggestions.
Minor comments:
Question 1: Title: "Nomogram and predict the overall survival ..." consider to change with "nomogram to predict overall survival"
Answer 1: Many thanks for your constructive comments and kind congratulation. The title was corrected.
Question 2: Abstract (Background):
"Performance was assessed by the concordance index (C-index)" please consider moving this sentence to the material and method paragraph of the abstract section
Answer 2: Many thanks for your great attention. The part was edited.
On behalf of all the authors,
Sincerely yours,
Atefeh Talebi, Ph.D.

Reviewer 3 Report
some of comments are not well justified .However over all the response to comments are well justified .
Author Response
Diane Jiang
Assistant Editor
International Journal of Environmental Research and Public Health
Dear Diane Jiang,
We thank you for the insightful editorial and review comments. We are pleased to be given the opportunity to submit a revised version of our manuscript.
We would like to thank the insightful editorial and reviewer for the constructive and competent criticism. We are very glad to hear that our manuscript has been accepted for publication in the International Journal of Environmental Research and Public Health.
According to the thoughtful comments of reviewers, we perform the reviewers ‘comments. Please find below our point-by-point replies to the reviewers’ comments, which are formatted in bold and numbered, followed by our reply in normal, green-coloured characters (our replies to the comments). Also, all changes in the manuscript are clearly marked by track change. The manuscript has sent in a separate word-processing file (Second revise manuscript with track changes). All authors have read and approved the revised version of the manuscript.
Reviewer #3:
Question 1: There are still certain typo mistakes and the quality of the images is still poor.
We look forward to hearing from you at your earliest convenience.
Answer 1: Many thanks for your helpful comment and kind congratulation. All authors and a native researcher edited all manuscript.
On behalf of all the authors,
Sincerely yours,
Atefeh Talebi, Ph.D.
